# Prognostic Model for Intracranial Progression after Stereotactic Radiosurgery: A Multicenter Validation Study

**DOI:** 10.3390/cancers14215186

**Published:** 2022-10-22

**Authors:** David J. Carpenter, Brahma Natarajan, Muzamil Arshad, Divya Natesan, Olivia Schultz, Michael J. Moravan, Charlotte Read, Kyle J. Lafata, Will Giles, Peter Fecci, Trey C. Mullikin, Zachary J. Reitman, John P. Kirkpatrick, Scott R. Floyd, Steven J. Chmura, Julian C. Hong, Joseph K. Salama

**Affiliations:** 1Department of Radiation Oncology, Duke University Medical Center, Durham, NC 27710, USA; 2Department of Medicine, University of Carolina Chapel Hill, Chapel Hill, NC 27599, USA; 3Department of Radiation Oncology, University of Chicago, Chicago, IL 60637, USA; 4Department of Radiation Oncology, University of Carolina Chapel Hill, Chapel Hill, NC 27599, USA; 5Radiation Oncology Clinical Service, Veterans Affairs St. Louis Health Care System, St. Louis, MO 63110, USA; 6Department of Radiology, Duke University Medical Center, Durham, NC 27710, USA; 7Department of Electrical and Computer Engineering, Duke University, Durham, NC 27710, USA; 8Department of Neurosurgery, Duke University Medical Center, Durham, NC 27710, USA; 9Department of Radiation Oncology, University of California San Francisco, San Francisco, CA 94143, USA; 10Bakar Computational Health Sciences Institute, University of California San Francisco, San Francisco, CA 94143, USA; 11Joint Program in Computational Precision Health, University of California San Francisco, San Francisco, CA 94143, USA, University of California Berkeley, Berkeley, CA 94720, USA; 12Radiation Oncology Clinical Service, Durham VA Health Care System, Durham, NC 27710, USA

**Keywords:** stereotactic radiosurgery, intracranial progression, brain metastases

## Abstract

**Simple Summary:**

To optimize surveillance for patients with brain metastases following stereotactic radiosurgery (SRS), we sought to validate a previously published nomogram estimating post-SRS intracranial progression (IP) risk. Among 890 patients completing an initial SRS course across two institutions 7/2017–12/2020, 53% were deemed high-risk for IP. Freedom from IP was superior for low-risk patients (*p* < 0.001), with a median of 13.9 months (95% CI 11.1–17.1 months) versus 7.6 months (95% CI 6.4–9.3 months) for high-risk patients. This large multisite cohort supports the use of an IP nomogram as a quick, simple means of stratifying patients into low- and high-risk groups for post-SRS IP.

**Abstract:**

Stereotactic radiosurgery (SRS) is a standard of care for many patients with brain metastases. To optimize post-SRS surveillance, this study aimed to validate a previously published nomogram predicting post-SRS intracranial progression (IP). We identified consecutive patients completing an initial course of SRS across two institutions between July 2017 and December 2020. Patients were classified as low- or high-risk for post-SRS IP per a previously published nomogram. Overall survival (OS) and freedom from IP (FFIP) were assessed via the Kaplan–Meier method. Assessment of parameters impacting FFIP was performed with univariable and multivariable Cox proportional hazard models. Among 890 patients, median follow-up was 9.8 months (95% CI 9.1–11.2 months). In total, 47% had NSCLC primary tumors, and 47% had oligometastatic disease (defined as ≤5 metastastic foci) at the time of SRS. Per the IP nomogram, 53% of patients were deemed high-risk. For low- and high-risk patients, median FFIP was 13.9 months (95% CI 11.1–17.1 months) and 7.6 months (95% CI 6.4–9.3 months), respectively, and FFIP was superior in low-risk patients (*p* < 0.0001). This large multisite BM cohort supports the use of an IP nomogram as a quick and simple means of stratifying patients into low- and high-risk groups for post-SRS IP.

## 1. Introduction

Approximately 20–40% of cancer patients develop brain metastases (BMs) [1,2]. Stereotactic radiosurgery (SRS), with or without surgical resection, is a standard of care for many patients with BMs [3]. To inform prognosis and guide treatment recommendations for patients with BMs, predictive models have long been used to estimate overall survival (OS) [4,5,6,7,8]. Such models, however, have generally not evaluated intracranial progression (IP) to inform management following intracranial therapy. Accurate IP estimation, particularly following SRS, may offer several unique benefits beyond those of OS. Moreover, as OS continues to improve for patients with BMs [9,10], IP has become an increasingly relevant clinical endpoint. Currently, for all patients completing SRS for BMs, the National Comprehensive Cancer Network recommends routine screening with magnetic resonance imaging every 2–3 months for 1–2 years, then every 4-6 months indefinitely [11]. However, these guidelines do not account for possible associations between time to post-SRS IP and patient-, tumor-, or treatment-specific variables. Performing post-SRS MRI brain surveillance at too short an interval may lead to unnecessary psychosocial and financial burdens; a surveillance interval too long increases the risk of symptomatic and/or diffuse intracranial progression.

To guide individualized post-SRS surveillance, an IP nomogram was developed from a large, multicenter cohort as a means of dichotomizing BM patients into low- and high-risk strata for time to post-SRS IP [12]. Prognostic factors from this nomogram include number of BMs (1, 2, ≥3), histology (melanoma vs. other), history of WBRT, and time from initial cancer diagnosis to diagnosis of any metastases (>5 yrs vs. ≤5 yrs). Potential limitations to this nomogram include a limited number of patients receiving immune checkpoint or molecularly targeted therapies and a lack of external validation. Herein, we examine IP nomogram performance for a multi-institutional cohort completing SRS after the date range included within Natarajan et al. [12], to assess IP risk in the context of contemporary multidisciplinary management patterns including immune checkpoint and molecularly targeted therapies.

## 2. Materials and Methods

For this institutional review board-approved retrospective analysis, we identified consecutive patients completing an initial SRS course for brain metastases at two institutions between July 2017 and December 2020. Single- and multi-fraction SRS cases were included, as were those with prior whole brain radiotherapy (WBRT) or surgical resection of brain metastases. Collected demographic variables and clinical characteristics included the following: institution, year of SRS completion, age, sex, race, Karnofsky performance status (KPS), primary tumor site, and sites of extracranial metastatic disease at time of SRS. Oligometastatic disease burden was defined as 1–5 metastases (i.e., non-locoregional) present across all anatomic locations, including intracranial disease at the time of SRS [13]. Exclusion criteria included age <18 years at time of SRS.

Systemic therapy agents and dates were manually obtained via chart review. Dates of initial cancer diagnosis were retrieved from pathology records, while dates of initial extracranial metastatic disease, initial intracranial disease, initial post-SRS intracranial progression (IP), and initial post-SRS extracranial progression (ECP) were determined via multidisciplinary clinical consensus per radiology reports, and, where available, pathology records. SRS treatment parameters obtained included number of irradiated BMs, number of SRS fractions (per patient, maximum number of fractions for any BM), and SRS dose. Per institutional protocol across both centers, post-SRS surveillance included MRI brain scans every 2–3 months at the discretion of the primary radiation oncologist. IP was defined as any clinical concern for distant and/or recurrent intracranial progression per multidisciplinary review of MR brain images. Patients were characterized as low- or high-risk for post-SRS IP per the nomogram proposed by Natarajan and colleagues (number of BMs [1, 2, ≥3], histology [melanoma vs. other], history of WBRT, and time from initial cancer diagnosis to diagnosis of any metastases [>5 yrs vs. ≤5 yrs]) [12]. Parameter distribution across low- and high-risk patients was assessed via Wilcoxon rank sum testing for continuous data and Chi-squared testing for categorical data. OS and freedom from IP (FFIP) were assessed with the Kaplan–Meier method. Both endpoints were analyzed from the time of SRS completion, with patients censored at the time of death for FFIP analysis. Assessment of parameters impacting FFIP was performed with univariable and multivariable Cox proportional hazard models, limiting multivariable models to parameters with α < 0.05 on univariable analysis. Patients with missing data were excluded from analysis. Data were collected and managed using REDCap electronic data capture tools, with all analyses performed using R Statistical Software (version 4.1.2; R Foundation for Statistical Computing, Vienna, Austria).

## 3. Results

We identified 890 patients completing SRS across two institutions between 2017 and 2020. As presented in Table 1, with reference values from Natarajan and colleagues [12], patients were a median age of 64 years, 55% female, and 73% Caucasian. In total, 71% of patients had a KPS of 80 or greater at time of SRS. Common primary tumor sites included non-small cell lung cancer (NSCLC; 47%), other (non-lung, breast, skin/melanoma, or renal; 18%), and breast (15%). 68% of patients had uncontrolled extracranial disease at time of SRS. Across six extracranial sites of interest (lymph nodes, lungs, bones, liver, adrenals, other), patients had a median of 2 involved sites at the time of SRS. Accounting for BMs as well as multiple metastases across extracranial sites, 47% of patients had oligometastatic disease.

Table 2 summarizes treatment characteristics for 2891 BMs across 890 patients in comparison to those reported by Natarajan and colleagues [12]. Prior WBRT and surgical resection were performed in 26% and 8% of patients, respectively. In total, 63% of patients completed any systemic therapy prior to SRS, including cytotoxic chemotherapy (51%), immunotherapy (28%), and molecularly targeted therapy (22%). Multi-fractionated SRS was used for at least one BM in 56% of cases, with single- and multi-fraction SRS doses ranging from 15–25 Gy and 18–35 Gy, respectively. Per the IP nomogram, 53% of patients were deemed high-risk (Table 3), primarily due to ≥2 non-melanoma BMs (50% of all patients) in the context of metastatic disease <5 years from cancer diagnosis (91%).

Median follow-up was 9.8 months (95% CI 9.1–11.2 months), with 304 (34%) patients alive at last follow up. For all patients, OS was 77.5% (95% CI 74.8–80.3%) at 3 months, 62.7% (95% CI 59.5–65.9%) at 6 months, 44.9% (95% CI 41.6–48.3%) at 12 months, 31.6% (95% CI 28.1–35.0%) at 24 months, 23.0% (95% CI 19.1–26.8%) at 36 months, and 19.2% (95% CI 14.8–23.6%) at 48 months (Figure 1A). IP nomogram high- versus low-risk classification was associated with an inferior OS (Figure 1B; HR 1.45, 95% CI 1.23–1.71; *p* < 0.001), with a median overall survival of 7.6 months (95% CI 6.4–9.3 months) for high-risk patients and 13.9 months (95% CI 11.0–17.1 months) for those at low risk.

Table 4 summarizes demographic, clinical, and treatment parameters across patients with low- versus high-risk IP nomogram classification. With respect to IP nomogram parameters, high-risk patients had a significantly greater number of brain metastases (96% multiple brain metastases vs. *p* < 0.01), as well as a smaller proportion of patients with a >5 year interval from initial cancer diagnosis to diagnosis of any metastatic disease (1% vs. 19%, *p* < 0.01). Overall distribution of primary tumor type was significantly different (*p* < 0.01) despite similar proportions with melanoma (7% vs. 9%), due to a greater proportion of other primary tumor types (19% vs. 11%) and fewer patients with breast primary tumors (11% vs. 20%) in the high risk cohort). Pre-SRS receipt of whole brain radiotherapy was not significantly different across groups (93% high-risk, 91% low-risk). Aside from IP nomogram parameters, significant differences across groups included controlled extracranial disease at time of SRS (27% high-risk, 39% low-risk, *p* < 0.01), oligometastatic disease at time of SRS (39% high-risk, 57% low-risk, *p* < 0.01), pre-SRS surgical resection (19% high-risk, 34% low-risk, *p* < 0.01), and pre-SRS chemotherapy receipt (45% high-risk, 57% low-risk, *p* < 0.01).

Multivariable Cox proportional hazard models (Table 5) showed significant associations between greater IP risk and the following parameters: skin/melanoma versus NSCLC primary tumors (HR 1.59, 95% CI 1.09–2.31), pre-SRS receipt of chemotherapy (HR 1.42, 95% 1.13–1.80), and number of irradiated BMs (2 vs. 1, HR 1.50, 95% CI 1.14–1.96; 3–5 vs. 1, HR 1.58, 95% CI 1.23–2.30; ≥6 vs. 1, HR 1.52, 95% CI 1.14–2.02).

FFIP was superior in low-risk patients (Figure 2, log-rank *p* < 0.0001). For low-risk patients, median FFIP was 10.7 months (95% CI 9.5–13.2 months), while FFIP was 96.0% (95% CI 94.1–98.0%) at 2 months, 87.4% (95% CI 84.1–90.9%) at 3 months, 71.5% (95% CI 66.8–76.6%) at 6 months, and 47.8% (95% CI 42.3–54.0%) at 12 months. For high-risk patients, median FFIP was 6.3 months (95% CI 5.7–7.3 months); FFIP was 88.6% (95% CI 85.6–91.7%) at 2 months, 75.2% (95% CI 71.0–79.6%) at 3 months, 52.8% (95% CI 47.7-58.4%) at 6 months, and 34.1% (95% CI 29.0–40.1%) at 12 months. Reference median FFIP values from the Natarajan validation cohort include 9.8 months (95% CI 6.6–14.9 months) for low-risk patients and 6.4 months (95% 5.3–10.0 months) for high-risk patients and [12].

## 4. Discussion

In this analysis, we tested the validity of a previously published nomogram to predict IP in a large cohort of almost 900 patients with BM treated with recently introduced systemic therapies. Validation of the Natarajan IP nomogram, which estimates post-SRS IP risk, was necessary as it was developed from a randomly assigned 2:1 training testing split across a multi-institutional cohort of 755 patients completing SRS between January 2002 and June 2017 [12]. In the current analysis, we demonstrate that while clinical outcomes continue to improve for patients completing SRS for BMs, this IP nomogram accurately discriminates between patients at high versus low risk for IP.

While other clinical prediction models for BM patients have historically prioritized OS as a primary endpoint to guide management of intracranial disease, we focused on IP as IP has become an increasingly relevant endpoint for a number of reasons. First, time to IP (e.g., intracranial control) has become increasingly independent of OS; advances in systemic therapies have drastically prolonged OS following BM diagnosis [9,10], exacerbated by lead-time bias in BM detection from improved MRI screening within cancer populations [5,14]. Additionally, for many patients with BMs, OS is improving due in some part to emerging systemic therapies with improved intracranial response rates. All together, these changes support the increased use of IP-free survival as a primary endpoint [15,16,17]. Second, for appropriately selected BM patients, radiotherapy practice patterns have shifted from WBRT to SRS in an effort to preserve neurocognition at the potential expense of distant IP risk [18,19,20,21]. Third, widespread implementation of response assessment in neuro-oncology (RANO) criteria has standardized IP radiologic assessment [22]. Finally, in contrast to OS, IP uniquely addresses individualized surveillance recommendations that may optimize early BM detection prior to symptomatic onset and/or diffuse dissemination, patient quality of life, and cost-effectiveness for patients and health systems. Accordingly, ongoing trials such as NRG BN009 (NCT04588246) are now incorporating IP (e.g., brain metastases velocity) into prospective trial design [23]. The present data provide valuable context for such efforts.

The contemporary cohort analyzed here is presumed to optimally incorporate recent clinical practices involving comprehensive molecular testing and administration of immune checkpoint inhibitor and molecular targeted therapies, and therefore is an optimal cohort to describe patterns of IP and test the validity of prior nomograms. While a limited number of nomograms have characterized post-SRS IP risk across distinct BM populations [12,24,25], associated external validation studies remain limited. As context for IP nomogram performance, we identified several distinctions between Natarajan and colleagues’ validation cohort and our large, multi-institutional cohort with respect to patient, tumor, and treatment characteristics [12]. Our cohort was 18% larger than that of Natarajan et al., and included more than twice as many BMs as well as a larger proportion of multi-fractionated SRS courses. Across both cohorts, a similar proportion of patients underwent surgical resection. In contrast, we observed lower rates of prior WBRT and prior chemotherapy, consistent with respective trends in increased SRS utilization [19] and increased MR brain screening [14]. Distributions across patient age, sex, race, and functional status were comparable. However, despite a comparable predominance of NSCLC patients, our cohort had smaller proportions of breast and melanoma patients, perhaps consistent with increasing efficacy of systemic therapies in these diseases. Natarajan and colleagues did not report frequencies of oligometastatic disease and pre-SRS receipt of immune checkpoint or molecularly targeted therapies [12]; however, we found no associations between these parameters and IP risk.

Following IP nomogram classification, patients differed with respect to a number of clinical parameters not included within the IP nomogram; namely, extracranial disease control, oligo- versus polymetastatic burden, prior surgical resection, pre-SRS chemotherapy, and primary tumor origins aside from melanoma. Of these parameters, only pre-SRS chemotherapy demonstrated significance on multivariable analysis of FFIP. The IP nomogram incorporates number of brain metastases, which carries direct implications for both metastatic disease burden (i.e., >5 combined intracranial and extracranial metastatic foci) and the clinical decision to pursue surgical resection. Receipt of pre-SRS chemotherapy would be expected to affect extracranial disease control. Together, these data suggest that while the IP nomogram appears to be an effective means of stratifying BM patients by high versus low IP risk, pre-SRS chemotherapy receipt warrants additional consideration.

For BM patients completing SRS in the context of contemporary multidisciplinary management, our data emphasize the importance of individualized prognostication accurately discriminating between patients with low and high IP risk. This differs from current recommendations for post-SRS intracranial surveillance, which are uniform across all patients with BMs [26,27]. Moreover, despite superior OS to that of the Natarajan cohort, we found corresponding median FFIP values for low- and high-risk patients to be within one month of those reported by Natarajan and colleagues [12]. Regarding clinical application, for patients at low risk for IP (e.g., fewer BMs, non-melanoma origin, prior WBRT, greater interval from cancer diagnosis to metastatic disease), extending the initial surveillance interval to 3 months appears justifiable using an approximate 10% clinical threshold for IP risk. However, for those at higher IP risk (e.g., more BMs, melanoma origin, no prior WBRT, metastatic disease within 5 years of initial diagnosis), an initial surveillance interval of 2 months or less may be appropriate to minimize the risk of symptomatic and/or diffuse intracranial progression when applying the same clinical threshold for IP.

Per IP nomogram stratification, low and high IP risk was also significantly associated with OS. Other nomograms and prognostic indices have long been used to supplement oncologists’ ability to accurately predict OS in patients with metastatic disease [28,29]. To address significant heterogeneity across outcomes for patients with BMs, initial prognostication tools such as the Radiation Therapy Oncology Group (RTOG) recursive partitioning analysis (RPA) model have assumed a foundational role in patient-specific intracranial management from RTOG 9508 onward [5,30]. Subsequent models have accounted for significant advances across multi-disciplinary BM management, as well as characteristics unique to specific primary tumor sites [4,6,7,31]. These revised models demonstrate significant improvements in both model discrimination (i.e., significant differences between strata) and individualized prognostication (i.e., absolute time to event estimation). More recently, brain metastases velocity has gained prominence as a validated OS prognostic tool at time of IP [32]; however, these models do not directly address IP risk following an initial SRS course as a primary outcome. Despite apparent overlap in clinical parameters associated with IP and OS, the IP nomogram appears best situated to guide post-SRS surveillance, distinct in clinical application from OS models that guide prognostication and related treatment decisions.

Limitations of this study include its retrospective scope, in which global improvements in clinical outcomes cannot be attributed to specific parameters. While, to our knowledge, this multicenter cohort represents the largest validation of post-SRS IP risk to date, patient numbers are relatively small in comparison to those of post-SRS OS models, particularly >1 year following SRS completion. FFIP, while more applicable to the subgroup of patients eligible for post-SRS imaging than IP-free survival, may underestimate true IP events, particularly among patients who expired within 3 months of SRS completion. Follow-up analyses are ongoing to address (1) limitations related to extrapolation of a NSCLC-predominant cohort to non-NSCLC populations through site-specific models, (2) uncertainty of correlation between OS and IP, and (3) optimal screening intervals for post-SRS surveillance MR brain imaging. However, the present analysis validates the use of the IP nomogram as a simple means of assessing high versus low IP risk following SRS in a large, multisite BM population that reflects contemporary utilization patterns for immune checkpoint and molecularly targeted therapies.

## 5. Conclusions

Given improvements across multi-disciplinary BM management, optimization of post-SRS surveillance through IP risk estimation has become increasingly important. This contemporary multi-institutional BM cohort supports the use of the IP nomogram as a simple means of stratifying patients into low- and high-risk groups for post-SRS IP to inform post-SRS surveillance.

## Figures and Tables

**Figure 1 cancers-14-05186-f001:**
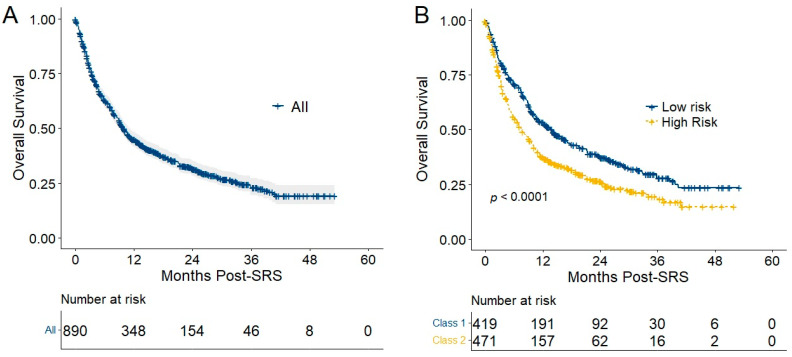
Overall survival following stereotactic radiosurgery is shown for all patients with 95% confidence intervals (**A**) and by intracranial progression nomogram classification (**B**). Abbreviation: SRS, stereotactic radiosurgery.

**Figure 2 cancers-14-05186-f002:**
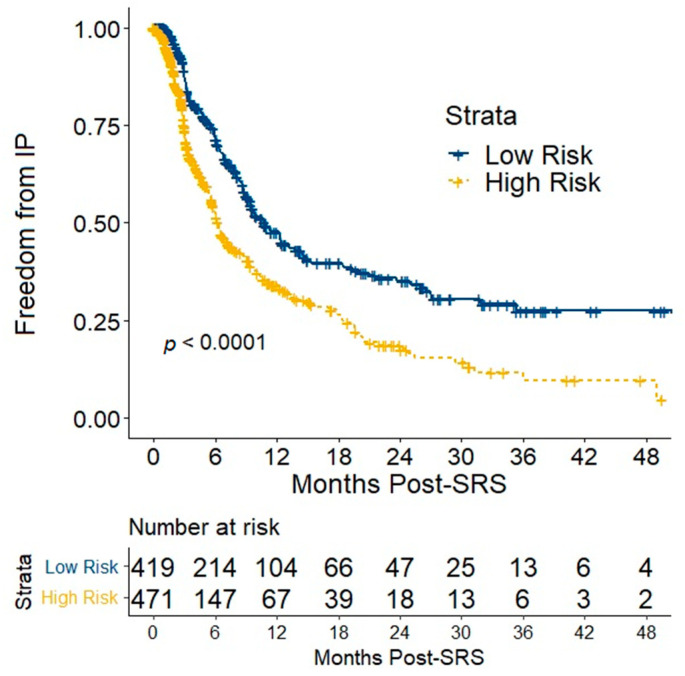
Freedom from intracranial progression (IP) is shown for all patients per IP nomogram risk stratification. Abbreviation: SRS, stereotactic radiosurgery.

**Table 1 cancers-14-05186-t001:** Patient demographics and disease characteristics across all patients. Abbreviations: Interquartile range, IQR; stereotactic radiosurgery, SRS; non-small cell lung cancer, NSCLC; small cell lung cancer, SCLC.

	Current Cohort (n = 890)	Natarajan 2019 (n = 755)
	N (%)	N (%)
**Institution**		-
1	162 (18)	
2	728 (82)	
**Year of SRS**		-
2020	266 (30)	
2019	260 (29)	
2018	265 (30)	
2017	99 (11)	
**Median age at SRS (range)**	64 (22–92)	60 (22–91)
**Sex**		
Female	491 (55)	441 (58)
Male	399 (45)	311 (41)
Unknown	0 (0)	3 (0.4)
**Race**		
White	653 (73)	566 (75)
Black	199 (22)	103 (14)
Other	38 (4)	13 (2)
Unreported	0 (0)	72 (10)
**Karnofsky Performance Status**		-
100	84 (9)	
90	308 (35)	
80	240 (27)	
70	153 (17)	
60	51 (6)	
50 or less	54 (6)	
**Primary Tumor Site**		
NSCLC	418 (47)	337 (45)
SCLC	60 (7)	-
Breast	137 (15)	146 (20)
Skin/Melanoma	69 (8)	129 (17)
Renal	43 (5)	54 (7)
Other	163 (18)	85 (11)
Unknown	0 (0)	3 (0.4)
**Extracranial disease at time of SRS**		
Controlled/None	289 (32)	304 (54)
Uncontrolled	601 (68)	309 (41)
Unknown	0 (0)	42 (6)
**Months from cancer diagnosis to initial metastases**		-
Median (range)	0.0 (0.0–449.2)	
Unknown	0 (0)	
**Months to extracranial disease**		
Median (range)	0.0 (0.0–449.2)	0.0 (range, 0.0–291.7)
Unknown	5 (0.6)	1 (0.1)
**Months to intracranial disease**		
Median (range)	11.4 (0.0–472.1)	14.5 (range, 0.0–291.7)
Unknown	0 (0)	6 (0.8)
**Number of involved extracranial sites at time of SRS**		-
Median (range)	2 (0–6)	
**Nodal metastases**	349 (39)	-
**Pulmonary metastases**	391 (44)	-
**Bone metastases**	318 (36)	-
**Hepatic metastases**	178 (20)	-
**Adrenal metastases**	98 (11)	-
**Other metastases**	112 (13)	-
**Metastatic burden at SRS**		-
Polymetastatic	468 (53)	
Oligometastatic	422 (47)	

**Table 2 cancers-14-05186-t002:** Treatment characteristics across all patients, with reference comparison to the cohort from [12]. Abbreviations: Interquartile range, IQR; stereotactic radiosurgery, SRS.

	Current Cohort (n = 890)	Natarajan 2019 (n = 755)
	N (%)	N (%)
**Prior surgical resection**		
Yes	232 (26)	176 (23)
No	658 (74)	579 (77)
**Prior whole brain radiotherapy**		
Yes	73 (8)	282 (37)
No	817 (92)	473 (63)
**Prior chemotherapy**		
Yes	450 (51)	513 (68)
No	440 (49)	227 (30)
Unknown	0 (0)	15 (2)
**Prior immunotherapy**		-
Yes	245 (28)	
No	645 (72)	
**Prior targeted therapy**		-
Yes	196 (22)	
No	694 (78)	
**Number of intracranial metastases treated with SRS**		
Total	2891	1407
Median (range)	2 (1–54)	1 (1–9)
**SRS fractionation**	*(per patient)*	*(per lesion)*
Single fraction	389 (44)	1297 (92)
Multi-fraction	508 (56)	103 (7)
2-fraction	39 (4)	1 (0.1)
3-fraction	68 (8)	21 (3)
4-fraction	15 (2)	0 (0)
5-fraction	379 (43)	81 (11)
Unknown	0 (0)	7 (0.5)
**Total SRS dose (Gy)**		
Single fraction median (range)	20 (15–25)	18 (5–25)
Multi-fraction median (range)	25 (18–35)	25 (12–35)

**Table 3 cancers-14-05186-t003:** Intracranial progression nomogram score distribution across all patients, with reference comparison to the test cohort from [12].

Nomogram Criteria	Current Report (n = 890)	Initial Testing Cohort (n = 248)
**Treated brain metastases (Melanoma)**		-
1 or 2: 35 points	39 (4)	
≥3: 100 points	28 (3)	
**Treated brain metastases (Non-Melanoma)**		-
1: 0 points	378 (42)	
≥2: 45 points	445 (50)	
**History of whole brain radiotherapy**		-
Yes: 0 points	73 (8)	
No: 15 points	817 (92)	
**Time from Cancer Diagnosis to Initial Metastases**		-
>5 years: 0 points	84 (9%)	
Within 5 years: 45 points	806 (91%)	
**Total points**		
0–85 points: Low Risk	419 (47%)	114 (46%)
≥86 points: High Risk	471 (53%)	134 (54%)

**Table 4 cancers-14-05186-t004:** Demographic, clinical, and treatment parameters across all patients by intracranial progression nomogram strata. Abbreviations: Interquartile range, IQR; stereotactic radiosurgery, SRS; planned target volume, PTV.

	Low Risk (n = 419)	High Risk (n = 471)	*p* Value
	N (%)	N (%)	
**Year of SRS**			0.89
2017	51 (12%)	48 (10%)	
2018	126 (30%)	139 (30%)	
2019	121 (29%)	139 (30%)	
2020	121 (29%)	145 (31%)	
**Median age at SRS (IQR)**	64.4 (56.3–72.1)	63.7 (54.6–72.0)	0.42
**Sex**			0.64
Female	241 (58%)	250 (53%)	
Male	178 (42%)	221 (47%)	
**Race**			0.35
White	306 (73%)	347 (74%)	
Black	101 (24%)	98 (21%)	
Other	12 (3%)	26 (6%)	
**Karnofsky performance status**			0.25
100–90	193 (46%)	199 (42%)	
80 or less	226 (54%)	272 (58%)	
**Primary Tumor Origin**			<0.01
Lung	213 (51%)	265 (56%)	
Breast	84 (20%)	53 (11%)	
Skin/Melanoma	39 (9%)	34 (7%)	
Renal	39 (9%)	31 (7%)	
Other	44 (11%)	88 (19%)	
**Extracranial disease at time of SRS**			<0.01
Uncontrolled	257 (61%)	344 (73%)	
Controlled/None	162 (39%)	127 (27%)	
**>5 years from cancer diagnosis to any metastases**			<0.01
No	339 (81%)	467 (99%)	
Yes	80 (19%)	4 (1%)	
**Metastatic burden at SRS**			<0.01
Oligometastatic	238 (57%)	184 (39%)	
Polymetastatic	181 (43%)	287 (61%)	
**Prior surgical resection**			<0.01
No	275 (66%)	383 (81%)	
Yes	144 (34%)	88 (19%)	
**Prior whole brain radiotherapy**			0.10
No	390 (93%)	427 (91%)	
Yes	29 (7%)	44 (9%)	
**Prior chemotherapy**			<0.01
No	180 (43%)	260 (55%)	
Yes	239 (57%)	211 (45%)	
**Prior immunotherapy**			0.18
No	287 (68%)	330 (70%)	
Yes	132 (32%)	141 (30%)	
**Prior targeted therapy**			0.28
No	388 (93%)	378 (80%)	
Yes	103 (25%)	93 (20%)	
**Number of intracranial metastases treated with SRS**			<0.01
1	381 (91%)	17 (4%)	
2	15 (4%)	147 (31%)	
3–5	14 (3%)	175 (37%)	
≥6	9 (2%)	132 (28%)	
**Median PTV of all brain metastases (IQR)**	7.0 (1.2–23.9)	8.6 (2.5–23.4)	0.59

**Table 5 cancers-14-05186-t005:** Univariable and multivariable analyses are provided for intracranial progression across all patients. Abbreviations: Interquartile range, IQR; stereotactic radiosurgery, SRS; non-small cell lung cancer, NSCLC; small cell lung cancer, SCLC.

	Univariate Analysis		Multivariate Analysis	
	HR (95% CI)	*p*	HR (95% CI)	*p*
**Institution**				
1	Ref			
2	0.83 (0.65–1.06)	0.14		
**Year of SRS**				
2017	Ref			
2018	1.13 (0.81–1.56)	0.47		
2019	1.08 (0.78–1.50)	0.64		
2020	0.98 (0.70–1.38)	0.93		
**Age at SRS, per year**	0.989 (0.982–0.996)	<0.01	0.993 (0.985–1.000)	0.06
**Sex**				
Female	Ref			
Male	0.95 (0.79–1.16)	0.63		
**Race**				
White	Ref			
Black	0.81 (0.64–1.02)	0.08		
Other	0.95 (0.59–1.52)	0.82		
**KPS**				
100–90	Ref			
80 or less	0.99 (0.82–1.19)	0.91		
**Primary Tumor Origin**				
NSCLC	Ref		Ref	
SCLC	1.27 (0.84–1.93)	0.26	1.03 (0.65–1.63)	0.91
Breast	1.21 (0.93–1.57)	0.15	0.96 (0.71–1.31)	0.81
Skin/Melanoma	1.73 (1.23–2.42)	<0.01	1.59 (1.09–2.31)	0.01
Renal	0.76 (0.46–1.25)	0.28	0.94 (0.56–1.57)	0.34
Other	1.23 (0.94–1.60)	0.12	1.15 (0.87–1.51)	0.12
**Extracranial disease at time of SRS**				
Uncontrolled	Ref			
Controlled/None	1.04 (0.86–1.26)	0.7		
**Months from cancer diagnosis to any metastases**	1.000 (0.998–1.002)	0.96		
**Months to intracranial disease**	1.000 (0.998–1.002)	0.81		
**Metastatic burden at SRS**				
Oligometastatic	Ref			
Polymetastatic	1.16 (0.96–1.40)	0.12		
**Prior surgical resection**				
No	Ref		Ref	
Yes	0.77 (0.62–0.95)	0.02	0.84 (0.67–1.05)	0.12
**Prior WBRT**				
No	Ref		Ref	
Yes	1.50 (1.08–2.09)	0.02	1.12 (0.77–1.62)	0.56
**Prior chemotherapy**				
No	Ref		Ref	
Yes	1.41 (1.17–1.70)	<0.01	1.42 (1.13–1.80)	<0.01
**Prior immunotherapy**				
No	Ref		Ref	
Yes	1.40 (1.21–1.85)	<0.01	1.21 (0.95–1.54)	0.12
**Prior targeted therapy**				
No	Ref			
Yes	1.06 (0.84–1.32)	0.64		
**Number of intracranial metastases treated with SRS**				
1	Ref		Ref	
2	1.65 (1.27–2.14)	<0.01	1.50 (1.14–1.96)	<0.01
3–5	1.70 (1.33–2.17)	<0.01	1.58 (1.23–2.03)	<0.01
≥6	1.64 (1.25–2.15)	<0.01	1.52 (1.14–2.02)	<0.01
**SRS fractionation**				
Single fraction	Ref			
Multi-fraction	0.95 (0.79–1.14)	0.57		

## Data Availability

Anonymized data are available upon request from the corresponding author.

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
