# Peer review of "Prognostic Model for Intracranial Progression after Stereotactic Radiosurgery: A Multicenter Validation Study"

_cancers, 2022, doi:10.3390/cancers14215186_

Round 1

Reviewer 1 Report

The work presented by Carpenter et al. aims to propose a nomogram for the stratification of patients with high- and low risks of intracranial progression after stereotactic radiosurgery. This nomogram was previously proposed by Natarajan and colleagues in 2019. The authors propose to assess this approach in a larger cohort (890 patients) across two institutions, taking into account the update in the standard of care of patients in recent years (targeted therapies, immunotherapy….). This retrospective study is interesting, the results are well-presented, and I agree with the authors that the ICP nomogram is useful for accurate discrimination between patients at high versus low risk for ICP.

I would suggest the authors remove lines 215-218 of the discussion, which seem inappropriate at this localization of the manuscript.

Author Response

The work presented by Carpenter et al. aims to propose a nomogram for the stratification of patients with high- and low risks of intracranial progression after stereotactic radiosurgery. This nomogram was previously proposed by Natarajan and colleagues in 2019. The authors propose to assess this approach in a larger cohort (890 patients) across two institutions, taking into account the update in the standard of care of patients in recent years (targeted therapies, immunotherapy….). This retrospective study is interesting, the results are well-presented, and I agree with the authors that the ICP nomogram is useful for accurate discrimination between patients at high versus low risk for ICP.

I would suggest the authors remove lines 215-218 of the discussion, which seem inappropriate at this localization of the manuscript.

Thank you for highlighting this. This template language has been removed.

Reviewer 2 Report

Estimating the outcome in patients with brain metastases is of great importance in radiosurgery practice and many efforts have been put forward to do so. Similarly, the authors aimed to validate a previously published nomogram for the estimation of post-SRS intracranial progression (ICP) risk, including number of BMs (1, 2, ≥3), histology (melanoma vs other), history of WBRT, and time from initial cancer diagnosis to diagnosis of any metastases (> 5 yrs vs ≤ 5 yrs). They grouped 890 patients into low and high-risk patients with this nomogram and found a superior freedom from ICP for low-risk patients (p<0.001). Although the effect of these parameters is already known, this is a well-written paper. But I will make some suggestions: 

1.     Although technically correct, I suggest the authors to use another abbreviation for intracranial progression. As ICP would confuse readers with intracranial pressure.

2.     What does “[JSM1]” stand for? (Page 3, Line 60)

3.     There are some technical errors with the text, such as inappropriate gaps between sentences, missing dots, etc.

4.     “Authors should discuss the results and how they can be interpreted from the perspective of previous studies and of the working hypotheses. The findings and their implications should be discussed in the broadest context possible. Future research directions may also be highlighted.” (Page 12, Lines 215-217) should be removed.

5.     “Data Availability Statement: In this section, please provide details regarding where data support-353 ing reported results can be found, including links to publicly archived datasets analyzed or gener-354 ated during the study. Please refer to suggested Data Availability Statements in section “MDPI Re-355 search Data Policies” at https://www.mdpi.com/ethics. If the study did not report any data, you 356 might add “Not applicable” here.” (Page 15, Lines 353-356) should be corrected.

6.     I think the title of the Table 5 should be corrected as “Univariable and multivariable analyses are provided for intracranial progression”, as hazard ratio with increased number of irradiated BMs increases, so the Table cannot demonstrate “freedom from intracranial progression”.

Author Response

Estimating the outcome in patients with brain metastases is of great importance in radiosurgery practice and many efforts have been put forward to do so. Similarly, the authors aimed to validate a previously published nomogram for the estimation of post-SRS intracranial progression (ICP) risk, including number of BMs (1, 2, ≥3), histology (melanoma vs other), history of WBRT, and time from initial cancer diagnosis to diagnosis of any metastases (> 5 yrs vs ≤ 5 yrs). They grouped 890 patients into low and high-risk patients with this nomogram and found a superior freedom from ICP for low-risk patients (p<0.001). Although the effect of these parameters is already known, this is a well-written paper. But I will make some suggestions: 

  1. Although technically correct, I suggest the authors to use another abbreviation for intracranial progression. As ICP would confuse readers with intracranial pressure.

Intracranial progression has been modified from ICP to IP throughout the document, including in the graphical abstract and figure 2.

  1. What does “[JSM1]” stand for? (Page 3, Line 60)

Thank you for highlighting this. This has been removed from the text.

  1. There are some technical errors with the text, such as inappropriate gaps between sentences, missing dots, etc.

Thank you for bringing this up. Paragraphs have been reformatted to standardize line breaks as well as a single space between sentences. While we believe this has been fully addressed, we welcome any additional noted technical errors.

  1. “Authors should discuss the results and how they can be interpreted from the perspective of previous studies and of the working hypotheses. The findings and their implications should be discussed in the broadest context possible. Future research directions may also be highlighted.” (Page 12, Lines 215-217) should be removed.

Thank you for highlighting this. This section has been deleted.

  1. “Data Availability Statement: In this section, please provide details regarding where data support-353 ing reported results can be found, including links to publicly archived datasets analyzed or gener-354 ated during the study. Please refer to suggested Data Availability Statements in section “MDPI Re-355 search Data Policies” at https://www.mdpi.com/ethics. If the study did not report any data, you 356 might add “Not applicable” here.” (Page 15, Lines 353-356) should be corrected.

Thank you for highlighting this. This section has been deleted.

  1. I think the title of the Table 5 should be corrected as “Univariable and multivariable analyses are provided for intracranial progression”, as hazard ratio with increased number of irradiated BMs increases, so the Table cannot demonstrate “freedom from intracranial progression”.

We agree with this suggestion and have changed the table 5 legend accordingly.
